# Effects of Condensed Tannins Supplementation on Animal Performance, Phylogenetic Microbial Changes, and In Vitro Methane Emissions in Steers Grazing Winter Wheat

**DOI:** 10.3390/ani11082391

**Published:** 2021-08-13

**Authors:** Byeng R. Min, William E. Pinchak, Michael E. Hume, Robin C. Anderson

**Affiliations:** 1Conservation and Production Research Laboratory, Agricultural Research Service, United States Department of Agriculture, Bushland, TX 79012, USA; bmin1@tuskegee.edu; 2Department of Agricultural and Environmental Sciences, Tuskegee University, Tuskegee, AL 36088, USA; 3Texas A&M AgriLife Center, P.O. Box 1658, Vernon, TX 76385, USA; 4Southern Plains Agricultural Research Center, Agricultural Research Service, United States Department of Agriculture, College Station, TX 77845, USA; michael.hume@usda.gov (M.E.H.); robin.anderson@usda.gov (R.C.A.)

**Keywords:** rumen, microbiome, condensed tannins, methane, similarity coefficient

## Abstract

**Simple Summary:**

Grazing wheat pasture is a common practice throughout the southeastern and south-central United States; however, the practice is limited by concerns regarding the occurrence of bloat. In addition, there are few reports concerning methane production by cattle grazing wheat pasture. Naturally occurring plant secondary compounds, including condensed tannins (CT), saponins, and essential oils, are extensively evaluated as natural alternatives to control bloat and to mitigate methane production. However, the effects of CT supplementation on ruminal gas production, rumen microflora community changes, and animal performance in stocker cattle grazing wheat forage are not fully defined. Supplementation with CT induced changes in ruminal bacteria, reduced methane emissions, and increased animal performance. These data indicate that CT supplementation may benefit stocker producers grazing wheat pasture by decreasing the incidence of bloat and increasing animal performance by changing rumen fermentation.

**Abstract:**

Eighteen growing rumen-cannulated steers, with initial body weight (BW) of 167.4 ± 7.10 kg, were randomly allocated to one of three treatments that included a control (0% CT) and two CT treatment levels (0.05% and 0.07% condensed tannins (CT)/kg BW) with two replicates each. Both in vivo and in vitro experiments were conducted. In Exp. 1, final BW and average daily gain were greater (*p* < 0.05) for the 0.07% CT treatment compared to either 0.05% CT or control groups. Rumen bacterial populations in steers fed winter wheat in the absence of CT represented large proportions of the moderate-guanines and cytosines (GC) containing bacterial clusters with similarity coefficient (SC) ranging from 64% to 92% In the presence of CT on day 0, day 20, and day 60, however, the SC was 60% or greater (90% SC) with multiple bacterial band clusters as shown by the denaturing gel gradient electrophoresis banding patterns. In Exp. 2, in vitro total gas, potential gas, and CH_4_ productions decreased (*p* < 0.01) as CT supplementation increased in steers grazing wheat forage. These results suggested that the administration of CT improved BW gain and induced bacterial community changes in the rumen of steers grazing wheat forage.

## 1. Introduction

Winter wheat (*Triticum aestivum* L.) is one of the most important food crops and is correspondingly the major source of winter forage supporting stocker cattle in the Southern Great Plains. However, wheat grazing promotes frothy bloat associated with high soluble crude protein (CP; 20–30% DM) levels, which is the major concern for grazing stocker cattle [1]. The incidence of frothy bloat in cattle grazing wheat pasture results from complex interactions among grazing management, forage, and individual grazing activity as modified by ambient environmental conditions [2,3,4]. Consequently, alternative frothy bloat control strategies, which improve protein flow from the rumen through the gastrointestinal tract and improve feed efficiency, are necessary. Supplementation of quebracho (*Schinopsis* spp.) condensed tannins (CT) is a means to decrease frothy bloat incidence and severity in stocker cattle [3,5] while increasing the protein flow from the rumen to abomasum, and thus could potentially modulate ruminal microbiota changes and aid in improving animal performance [6,7]. Condensed tannins, also known as proanthocyanidins, are oligomers or polymers of flavan-3-ol units (e.g., catechin subunits; [8]), which have been used within ruminant ecosystems as a feed additive. Tannins are thought to have both beneficial or detrimental effects on dietary feeding value and animal performance [7,9,10]. Although there are limitations of the techniques to be applied, CT is thought to reduce CH_4_ production and improve animal performance [11,12]. However, high amounts of CT in the diet (>5% CT DM) can adversely affect animal performance when dietary protein is a limiting component, largely due to reductions in the absorption of amino acids in the gastrointestinal tract [10,13]. Biologically occurring plant CT is widely assessed as a natural alternative to influence bloat and to lessen methane production. However, the effects of CT supplementation on ruminal gas production, rumen microflora community changes, and animal performance in stocker cattle grazing wheat forage are not well-defined.

The PCR-based denaturing gradient gel electrophoresis (DGGE), while not a new technique, makes it possible to visualize PCR products (genomic guanines (G) and cytosines (C; GC) content) representing predominant microbiota dynamics elicited by a given diet [14,15]. Therefore, the DGGE banding pattern illustrates the dominant bacterial molecular fingerprinting that facilitates profiling and comparison of major microbial diversity changes between gastrointestinal samples. The genomic GC content, i.e., the proportion of G and C among all nucleotides in the genome, is one of the most frequently used taxonomic markers in microbiology [16,17,18,19,20,21]. Based on the GC content, various phyla could be characterized by different means of GC values [17]. A phylum of Firmicutes, Gram-positive bacteria that are grouped based on their low-low GC content [22,23,24,25,26]. In contrast, the phylum of Bacteroidetes is composed of bacteria that are generally Gram-negative, with a high GC content [27]. Therefore, GC content may be given for a certain fragment of genome DNA or RNA as a qualifying characteristic [19,28,29,30]. Recently, Min et al. [28] reported that there were linear increases in percentages of Firmicutes associated with increasing average daily gains (ADG) in meat goats grazing simple and mixed pastures. In addition, the ratio of Firmicutes to Bacteroidetes (F/B) in the rumen may increase when tannin-containing diets are included in the mixed ration [22,24,29]. Therefore, the relative abundance examination of GC content associated with Firmicutes [22,25,26] and Bacteroidetes [27] phylum abundance will be useful for exploring the structure of gut microbiota as an estimate of rumen fermentation and animal performance [28,29]. In the present study, DGGE was used as the method to visually profile and monitor changes occurring in various microbial communities according to a percentage similarity coefficient (% SC) and GC content. Despite years of research on CT supplementation in ruminant diets, the practical (e.g., on-farm) impacts of supplementation of CT on animal performance, rumen microbiome changes, and enteric methane (CH_4_) emission are still poorly understood in stocker cattle grazing winter wheat forage, especially as the trial may not be able to resolve all the question which arise. Our primary objective of Exp. 1 was to quantify the effect of CT supplementation on animal performance and microbial community changes of steers grazing winter wheat forage. A second in vitro experiment was to evaluate the effects of ruminal fluid from eighteen cannulated steers fed different dosages of quebracho CT (0%, 0.05%, and 0.07% CT/kg of body weight (BW)) on ruminal gas and CH_4_ production.

## 2. Materials and Methods

### 2.1. Experimental Design

The experiment was conducted at Texas A&M AgriLife Research and Extension Center, Vernon, TX. The Texas A&M University Animal Care and Use Committee approved the animal care, handling, and sampling procedures used during this experiment (Texas A&M IACUC AUP 04105). Grazing in vivo experiment (Exp. 1) was designed to quantify the effect of CT supplementation on average daily gain (ADG) and phylogenic rumen microbiome changes in steers grazing winter wheat. Eighteen growing rumen-cannulated steers (Angus; 167.4 ± 7.10 BW kg) were blocked by initial BW and were randomly allocated to 1 of 3 treatments that included a control (0% CT; warm water infusion) and 2 CT treatment levels (0.05% and 0.07% CT/kg BW) with two replicates each to enumerate animal performance and rumen microbiota profiles in steers grazing winter wheat forage (C3 annual grass; February to April) diets up to 60 d after a 4 w adaptation period at the Smith–Walker Research Unit, Vernon, Texas. Animal BW changes were measured on day 0, day 10, day 20, day 42, day 50, and day 60 to ADG for 60 d. A second in vitro experiment was conducted to enumerate the effect of ruminal fluid from steers fed quebracho CT (0, 0.05, and 0.07% CT/kg of BW) on in vitro gas and CH_4_ production from minced fresh wheat forage incubated with ruminal inoculums obtained from cannulated steers (*n* = 6 steers/treatment) grazing winter wheat forage.

### 2.2. Experiment 1 (In Vivo Experiment)

Steers were transferred to two replicate wheat pastures (4.7 ha/paddock) per treatment (*n* = 6) from 17 February to 17 April and allowed to graze for up to 60 days. A pure sward of winter wheat forage was used under continuous grazing during the experimental period. All experimental animals were offered ad libitum a free choice of mineral supplement (ACCO Wheat Advantage Mineral, Minneapolis, MN, USA) and water. The hand-plucked forage samples (*n* = 3) were collected twice (7 March and 17 April) by hand-clipping above ground (soil level) level. These forage samples were equally composited (200g/sample) and stored at −20 °C until used for chemical composition analysis and in vitro ruminal gas analysis. Rumen samples were collected 2 h post-infusion (at 10:30 to 11:30 a.m.) from steers. Rumen contents (~500 g/steer) from individual steers were obtained from 18 cannulated steers fed winter wheat forage on d 0, d 20, and d 60, without or with CT supplemented groups. All forage samples were positioned on ice and immediately transferred to the laboratory and frozen at −70 °C until DNA extraction.

#### Condensed Tannins Preparation and Supplementation

Two levels of quebracho CT (0.05% and 0.07% CT/kg of BW) and control (0% CT; warm water only) treatment were premixed with warm water (approximately 30 °C) and introduced once daily (0800) via rumen cannula of steers grazing wheat forage as a main source of diet. The amount of CT extract supplementations was then injected into the rumen of cannula adjusted by changing over time depending on an individual animal’s BW growth. Animals were weighed at 10 day intervals before administration to obtain a precise recording of the animal’s BW. Water-soluble quebracho CT extract was used as the source for CT (99% solubility; Chemtan Company, Inc., Exeter, NH, USA). This type of quebracho CT extract contains about 75% CT/DM and a small number of simple phenols. Structurally, quebracho CT possesses 4.5 to 8.5 polymer units of the quebracho-catechin, with a range of molecular weight from 1978 to 3878 [31].

### 2.3. Experiment 2

#### Effect of Ruminal Fluid on In Vitro Gas and Methane Production in Steers Fed Condensed Tannins

For in vitro gas analysis, rumen fluids were collected (~300 g/steer) on d 60 from 18 cannulated steers to 2 h post-infusion of CT or warm water (control) infusions that were grazing winter wheat forage. These rumen fluids were employed into pre-warmed insulated Thermo bottles, transported to the laboratory, homogenized, filtered through four layers of cheesecloth, equally pooled (50:50) by each treatment, and used as an in vitro inoculum. Rumen fluid was kept in a water bath at 39 °C with CO_2_ saturation until inoculation.

Before running the incubations, frozen samples of winter wheat forage were chopped to 2 to 3 cm lengths, and minced (2–3 mm particle size) using a blender (blender Model DS-7, Warning Products Co., Winsted, CT, USA) to obtain a particle size similar to that of chewed forages [32]. Freshly minced wheat forage as a substrate was added (5 g of fresh weight; 1.725 g DM) into each fermentation jar for in vitro incubation. Sub-samples of minced wheat forage (50 g of fresh weight) were oven-dried at 65 °C to a constant weight (>48 h) then were ground small enough to pass through a 1 mm mesh sieve for laboratory analysis. Triplicated in vitro ruminal gas production was measured as plunger displacement (cc) at 0, 2, 4, 6, 8, and 12 h incubation periods [32]. In vitro incubation temperature and time of incubation were chosen at 39 °C for 12 h periods in the present study, as Paisley and Horn [33] and Min et al. [34] indicated that the most in vitro gas production and CH_4_ emissions occurred within 8 to 12 h at temperatures for fresh winter wheat forage.

Flask stoppers were furnished with rubber tubing linked to 60 mL syringes (Tyco Health Care Ltd., Mansfield, MA, USA). Ruminal gas production was determined from an in vitro rumen incubation procedure in which 5 g of composited minced fresh hand-plucked wheat forage harvested from 7 March to 17 April was employed in 250 mL volumetric flasks containing 20 mL of ruminal fluid, 30 mL of artificial saliva, pH 6.8 which was saturated with CO_2_ gas and held at 39 °C [35,36]. All syringes were greased with a dose of syringe oil (Jupiter Vet products; Harrisburg, PA, USA) to assure reliable plunger resistance and movement. Cumulative ruminal gas production was recorded for 12 h, which records pressure build-up in individual vessels. After incubating for 12 h, one 10 mL aliquot of collected gas samples was taken from each syringe for CH_4_ concentration analysis using a SABLE gas analyzing system (Sable Systems International, Las Vegas, NV, USA).

### 2.4. DNA Extraction and Denaturing Gradient Gel Electrophoresis

Changes in predominant microbial populations are revealed by DGGE as the presence or absence of amplicon bands. The extent of PCR amplicon migration into the polyacrylamide gel urea-formamide denaturing gradient gel and melting (separation of the double-stranded DNA) domains is determined by the unique GC contents, base-pair primary sequences, and interactions between associated bases [37]. Genomic bacterial DNA was separated from 0.5 mL of each rumen fluid sample according to the method described in the QIAamp DNA Mini Kit (QIAGEN, Valencia, CA, USA) and DNA concentration was measured with a GeneQuant Pro (Biocompare, Valencia, CA, USA). Denaturing gradient gel electrophoresis was run according to the method of Muyzer et al. [37], with the modification of using bacteria-specific PCR primers to conserved regions flanking the variable V3 region of 16S rDNA genes. Primers (50 pmoL of each per reaction mixture; primer 2, and primer 3 (Integrated DNA Technologies, Inc., Coralville, IA, USA)) with a 40-based GC clamp [37,38] were mixed with Jump Start Red-Taq Ready Mix (Sigma Chemical Company, St. Louis, MO, USA), according to the kit instructions, 250 ng of template DNA from rumen digesta of individual steers, and 5% (wt/vol) acetamide to eliminate preferential annealing [39]. Amplifications were on a PTC-200 Peltier Thermal Cycler (MJ Research Inc., Waltham, MA, USA) with the following program: (1) denaturation at 94.9 °C for 2 min; (2) subsequent denaturing at 94 °C for 1 min; (3) annealing at 67 °C for 45 s, −0.5 °C per cycle [40,41]; (4) extension at 72 °C for 2 min; (5) repeat steps 2 to 4 for 17 cycles; (6) denaturation at 94 °C for 1 min; (7) annealing at 58 °C for 45 s; (8) repeat steps 6 and 7 for 12 cycles; (9) extension at 72 °C for 7 min; and (10) 4 °C final.

Polyacrylamide gels (8% (vol/vol) acrylamide-bisacrylamide ratio 37.5:1; Bio-Rad Laboratory, Richmond, CA, USA) were cast with a 35% to 60% urea-deionized formamide (USA Amersham Life Sciences, Cleveland, OH, USA) gradient; 100% denaturing acrylamide was 7 M urea and 40% deionized formamide. Amplified samples were integrated with an equal volume of 2 times concentrate loading buffer (0.05% (wt/vol) bromophenol blue, 0.05% (wt/vol) xylene cyanol, and 70% (vol/vol) glycerol) and 4 µL were placed in each sample well (16 well comb). Gels were placed in a DCode Universal Mutation Detection System (Bio-Rad) for electrophoresis in 0.5 time concentrate TAE (20 mM Tris (pH 7.4), 10 mM sodium acetate, 0.5 M EDTA) at 59 °C for 17 h at 60 V. Gels were stained with SYBR Green I (1:10,000 dilution; Sigma), and fragment pattern relatedness was determined with Molecular Analysis Fingerprinting Software, Version 1.6 (Bio-Rad Laboratory, Hercules, CA, USA) based on the Dice percentage similarity coefficient and the unweighted pair group method using arithmetic averages (UPGMA) for clustering.

### 2.5. Chemical Analysis

Total crude protein (CP) from fresh forage samples was determined by the Kjeldahl digestion procedure [42]. The neutral detergent fiber (NDF), acid detergent fiber (ADF), and in vitro dry matter digestibility (IVDMD) of dried forage samples (24 h) were determined using the Filter Bag Technique (ANKOM Technology Corp., Fairport, NY, USA).

### 2.6. Statistical Analysis

In vivo (Exp. 1) and in vitro (Exp. 2) data were analyzed with Proc GLM procedures of SAS (SAS, 1987). Treatment means were separated by the least significant differences when overall F-values were significant (*p* < 0.05). The main effect means for dietary treatments (CT effect) and time were reported in tables as there were no CT by time or replicates interactions for animal performance and DNA concentration (*p* > 0.05). Replications were the experimental unit and were treated as a random effect. Data are presented as least square mean values with the SEM. Variables in Exp. 1 included DNA concentration, animal BW changes, and ADG in steers grazing winter wheat forage. Initial body weight from day 0 was used as a covariate for BW changes. The model included diets, sampling time, and levels of tannins supplementation. The variables in Exp. 2 included in vitro ruminal gas, potential gas, and CH_4_ productions.

In vitro cumulative gas production (Y) parameters (Exp. 2) were calculated using the exponential equation of Orskov and McDonald [43].
Y = a + b(1 − e-ct)
where Y was expressed as gas production in time t; a, b, and c being constants of the exponential equation where a = the gas production at time 0, b = the gas production during the time (t), and c = the gas rate of the ‘b’ fraction. The constants b and c for each treatment were computed with the method described by Min et al. [32] using the Non-Linear Regression (NLIN) procedure from SAS (1987). Microbial clusters (groups) are verified by sequentially comparing the patterns and the construction of a relatedness tree (dendrogram) considering the relative similarities coefficient (% SC; Exp. 1). The amount of similarity is revealed by the relative closeness or grouping and is indicated by the %SC bar located above each dendrogram [14].

## 3. Results

### 3.1. Chemical Composition of Forages

The chemical composition of winter wheat forage is shown in Table 1. Across the grazing season, total CP and IVDMD were greater (*p* < 0.001) for the growth stage (7 March) than for the boot stage (7 April), whereas NDF content was greater (*p* < 0.01) for the boot stage than the growth stage with advanced plant development.

### 3.2. Experiment 1: Effects of Condensed Tannins on Animal Performance

Animal BW changes and ADG in steers grazing winter wheat forage in response to CT supplementation are shown in Figure 1 and Table 2. Although initial BW of steers was covariate by day 0 among treatment groups (Figure 1), final BW and ADG (*p* < 0.04) were greater for the 0.07% CT treatment group compared to either 0.05% CT or control groups (Table 2). There was no significant interaction between CT treatment by time of grazing interaction for animal BW changes (*p* > 0.05).

### 3.3. DNA Concentration and Ruminal Microbiota Changes

Genomic DNA concentrations are presented in Table 3. Ruminal genomic DNA concentrations in steers fed winter wheat forage on day 20 were similar, while DNA concentrations tended to be higher (*p* = 0.06) for CT groups than for the control diet on day 60 (Table 3). An average DNA concentration was greater (*p* < 0.04) for the CT groups than for the control. There was no significant interaction between CT treatment by time of grazing interaction for DNA concentration (*p* > 0.05).

Dendrograms of 16S rDNA V3 region amplicons from the rumen of steers grazing winter wheat forage with or without CT supplementation are presented in Figure 2, Figure 3 and Figure 4. Rumen bacterial populations in cannulated steers grazing winter wheat forage diets in the absence of CT supplementation represented large proportions of the moderate-GC containing bacterial clusters with similarity ranging from 64% to 92% SC (Figure 2). In the presence of CT on day 0, day 20, and day 60, however, the similarity was 60% or greater (90% SC) with multiple bacterial band clusters as shown by the DGGE banding patterns, except for steer #11 (<61% SC) on day 20 (Figure 3). Multiple bacterial band clusters were common in all 1% CT samples, while one or two major band clusters were common in steers fed winter wheat pasture in the absence of CT. Likewise, multiple banding patterns were seen in steers fed 0.07% CT supplementation, with 62% to 92% SC, but increasingly low-GC-containing bacterial communities were present (Figure 4), except for steers #16 and #18, compared to either 0.05% CT (Figure 3) or wheat forage diet (Figure 2). One or two major bands were common in the winter wheat diet, but the bands were found in multiple combinations of the DGGE banding patterns with high proportions of low-GC bands in CT groups. Across the diets, the banding patterns among replications (between paddocks) varied from 50% to 90% SC.

### 3.4. Experiment 2: In Vitro Gas and Methane Production in Rumen Fluid from Steers Fed Condensed Tannins

In vitro rumen fluid gas and CH_4_ production from cannulated steers fed CT supplementation are shown in Table 4. Although the mean rate of ruminal gas production, c (mL/h) was numerically lower for CT-supplemented groups than for the control group, total gas (mL/g DM), gas potential (a + b), and CH_4_ (mg/g DM) production decreased (*p* < 0.001) by 38%, 32%, and 70.8%, respectively, with increasing rates of CT supplementation in steers grazing winter wheat forage.

## 4. Discussion

The principal objective of this study was to determine in vitro ruminal CH_4_ production, in vivo animal performance, and phylogenic microbial community changes associated with different levels of CT supplementation in steers grazing winter wheat forage. The main findings showed that CT supplementation reduced in vitro CH_4_ production while increasing ADG. These outcomes occurred principally through reducing rumen gas production and altering microbial community changes. Collectively, these results suggest that CT supplementation has the potential to effectively mitigate CH_4_ emissions and to improve animal performance without the negative effects associated with stocker cattle grazing winter wheat forage. The results also indicate that dietary CT-induced animal performance and ruminal microbial community changes have the potential to reduce CH_4_ production.

### 4.1. Exp. 1: Animal Performance

Tannin-rich diets can alter performance and health in ruminants. Research on animal performance (e.g., ADG and milk production) associated with tannin-rich diets or CT extracts in vivo have been reported [7,9,11,44,45,46]. In addition to the impact on ruminal fermentation profiles [47], CT-rich diets fed to ruminants can have beneficial effects on nutrition, reproductive rates [48], bloat mitigation [34,49,50], gastrointestinal nematodes control [51], and improve milk quality through increasing concentrations of unsaturated fatty acids [47]. In the present study, positive responses with CT addition (0%, 0.05%, or 0.07% CT/kg BW) for ADG in steers grazing winter wheat suggest that it may be possible to have an optimum CP:CT ratio in the diet to improve animal weight gain [9,52]. This observation agrees with the results of a sheep CT-containing diet study that showed the optimum level of tannins in sheep diets, as measured by animal responses, was 2% to 3% CT DM [13,49]. Several studies reported that dietary CP:CT ratios were one of the most important factors on ADG in meat goats, grazing steers, and feedlot cattle [5,24,52,53]. In contrast, it has been reported that tannin levels had little effect on ADG and carcass characteristics in feedlot cattle [46], possibly due to a low CP:CT ratio in the diet [52]. Negative effects have also been reported, stating that the high dose level of quebracho CT (0 vs. 166 g/kg of DMI) impairs fiber digestion and induces toxicosis in sheep [54]. This difference was probably due to greater sensitivities of diet preferences [13], dietary CP levels (CP:CT ratio [12,52,55]), and microbial community diversity [28,29]. Therefore, plant tannins could be a useful tool as a potential growth promotor for improving performance when dietary protein in cattle diets is not a limiting factor. Further research is needed to assure a sustainable supply of abundant and safe food and other livestock products and improving economic profitability while reducing detrimental effects.

### 4.2. Rumen Bacterial Diversity

Interestingly, microbial genomic DNA concentrations were higher for CT groups than for the control in steers grazing winter wheat, regardless of time. It is likely that increased rumen microbial DNA biomass, when diets contained additions of CT up to 0.05% or 0.07% CT/kg of BW, increased the supply of microbial protein (17.6%) for animal growth (Table 3). Similarly, it appears that Firmicutes and Bacteroidetes phyla populations gradually increased with increasing CT-rich peanut skin diet (0.08 to 1.05 CT/DM) by incubation with 30% wet distillers’ grains plus solubles [29]. These findings indicate that increasing CT concentrations up to 1–3% CT/kg of DMI increased the amount of by-pass protein from the rumen, without reducing the amount of microbial protein synthesized [32,56,57]. McNeill et al. [58] reported that CT in mimosa legume forage (*Leucaena leucocephala*) increased the by-pass protein, while did not affect the efficiency of rumen microbial protein synthesis. These results indicate that each steer sample had its unique profile with wide ranges of animal-to-animal variation.

The GC content of bacterial genomes ranges from 16 to 75%, and extensive ranges of genomic GC content are observed, including both Gram-negative and Gram-positive bacteria [17]. Tajima et al. [23] and Edwards et al. [59] reported that the GC patterns were found in multiple GC banding patterns in cannulated dairy cows fed a mixed diet, with low-GC Gram-positive bacteria (generally Firmicutes; 52 to 54%), Gram-negative Bacteroidetes types (38 to 40%), Gram-negative Spirochetes (2.4%), and intermediate GC Gram-negative Proteobacteria (4.7%). Recently, 16S sequencing data revealed that predominant bacterial phyla were Firmicutes (65%), Actinobacteria (13%), Bacteroidetes (10%), and Proteobacteria (5%) across all samples in steers grazing winter wheat forage, [1] which are similar to other reports on the rumen microbiome [5,53,60].

Rumen bacterial populations in cannulated steers grazing winter wheat forage diets in the absence of CT represented large proportions of the moderate-GC containing bacterial clusters, with similarity ranging from 64% to 92% SC (Figure 2). Various bacterial band clusters were prevalent in all 0.05% CT samples, while, in the absence of CT, one or two key band clusters were mutual in steers fed winter wheat pasture. Additionally, in the presence of CT on day 0, day 20, and day 60, the similarity was 60% or more SC (90% SC) with numerous bacterial band clusters, except for steer #11 (<61% SC) on day 20 (Figure 3). Pitta et al. [61] reported that the predominant genera were Prevotella (up to 56%) genus in steers grazing wheat forage with less than 90% similarity. Likewise, multiple banding patterns were seen in steers fed 0.07% CT supplementation, with 62% to 92% SC, but increasingly low-GC-containing bacterial communities were present, suggesting that the supplementation of CT induced bacterial community changes (probably more Firmicutes). Pitta et al. [61] reported that genera from Firmicutes such as Clostridium, Eubacterium, and Butyrivibrio increased, while Prevotella from Bacteroidetes decreased in bloated samples. According to Carrasco et al. [62], cannulated Holstein steers supplemented with tannins (quebracho and chestnut tannins; 2g/kg of feed) in a total mixed ration presented a higher Firmicutes to Bacteroidetes ratio in comparison with the control group. A recent study found that the richness of Firmicutes in the rumen beneficially associates with the ADG in steers and meat goats, suggesting that these bacteria play a significant role in the feed efficiency of ruminants [28,63]. Even though the increase in Firmicutes to Bacteroidetes ratio influenced by tannins can improve animal performance in ruminants [29,64], the current study did not classify the bacterial phylum levels and needed further study.

Tannins are generally regarded as inhibitory to the growth of microorganisms [65,66,67], but the mechanisms and dynamics of rumen bacterial ecosystems are poorly understood. Growth of ruminal bacteria (*Butyrivibrio fibrisolvens*, *Eubacterium* spp., *Ruminobacter amylophilus*, and *Streptococcus bovis*) was reduced by CT supplementation, but a strain of *Prevotella ruminicola* was tolerant to CT (<400–600 g/mL of CT) from sainfoin (*Onobrychris viciifolia*; [68]) and birdsfoot trefoil (*Lotus corniculatus*; [69]) diets. Min et al. [70] reported that *Fibrobacter succinogenes*, *S. bovis*, and *Prevotella ruminicola* strains were relatively stable with time (day 0, day 10, and day 25) in the rumen of steers not receiving CT supplementation. However, with supplementation of chestnut and mimosa tannin extracts, populations of *F. succinogenes* and *S. bovis* increased with time, while *Eubacterium ruminantium* was not consistently detected over the grazing period. Several species, including *S. gallolyticus*, *Clostridium* sp., and Proteobacteria, have been identified as tannin-tolerant bacteria [71,72]. In the present study, the GC patterns were found in multiple GC banding patterns in steers fed 0.05% or 0.07% CT supplementation, with 60% to 92% SC, but increasing low-GC containing bacterial community (Figure 3 and Figure 4) compared to wheat forage diet (Figure 1), suggesting that the administration of CT induced bacterial community changes [24,62].

The rumen microbial populations of steers may also be dependent upon available nutrients in the CT-containing diet, as well as on tolerance to CT per se [65]. In grass-fed sheep, clusters of the low-GC-containing Gram-positive bacteria predominated (74.2%), whereas, in the tannins-containing mulga tannins (*Acasia aneura*; 5–24% CT/DM)-fed sheep, 78% of clusters were in the Cytophaga-Flexibacter-Bacteroides genera [73]. Furthermore, Min et al. [66] reported that when the sheep diets were changed from perennial ryegrass/white clover (control) to 3% CT-containing forage (*L. corniculatus*), populations of the proteolytic rumen bacteria *Clostridium proteoclasticum* B316 (68%), *Eubacterium* sp. C12 (44%), *S. bovis* B315 (40%), and *B. fibrisolvens* C211a (16%) were decreased. In contrast, Min et al. [74] reported that when meat goats received CT-containing ground pine bark (2 to 4% CT DM) up to 30% DM in a grain mixed diet, Firmicutes were significantly increased, while phylum Bacteroidetes populations were decreased. Possible reasons for these differences are that different antimicrobial activities of CT and different types of tannins have been reported [67,75]. These data also seem to suggest that CT binding capacity, molecular weight, and tolerance may be correlated [71,76]. The shift in GC content in rumen bacterial community changes indicates that the growth and inhibiting effects of CT on rumen bacteria are dependent on CT type or amount [10]. At a concentration of 0.07% of kg BW, CT was more likely to induce low-GC-containing bacterial strains compared to the 0.05% CT treatment group, indicating that dietary CT modulated ruminal bacterial community changes. Quebracho CT is potentially a value-added supplement that can increase ADG in stocker cattle wheat pasture systems. The mechanism by which CT inhibits some rumen bacteria, but not others, is poorly understood and needs further study.

### 4.3. Exp. 2: Condensed Tannins and In Vitro Ruminal Methane Production

Mixed rumen fluids (liquid only) were used for in vitro study as Pitta et al. [61] reported that wheat rumen contents were more homogenized without a fibrous material present, compared to bermudagrass (*Cynodon dactylon*) hay. The same authors found that below the 0.9% threshold of the plotted data from whole and liquid wheat fractions for the Clostridium and Bacteroides populations, while predominant genera were Prevotella (up to 56%) genus on the wheat diet regardless of the all fractions of wheat. A possible reason for the increase in Prevotella genus on all fractions of wheat is related to the homogenized nature of rumen contents with less stratification of fiber and greater rumen fluid content [61] than has been reported in the literature [77,78].

Methane emissions from ruminal fermentation of dietary feedstuffs are a significant source of agricultural CH_4_ emission. Regardless of the production system adopted, ruminant livestock contributes to greenhouse gas (GHG) emissions associated with climate change. Among the diverse dietary approaches examined to decrease methanogenesis in ruminants, the use of CT shows potential, but has received only uncertain consideration [10,79]. The present study shows that quebracho CT in vitro in rumen fluid exhibited lower ruminal gas (38%) and CH_4_ production (71%) compared to steer diets that did not contain CT, across the winter wheat forage diets. These results are consistent with those of McMahon et al. [80], who reported that CH_4_ emissions decreased linearly with increasing CT-containing (11.3% CT) forage (sainfoin; *Onobrychis viccifoli*) supplementation compared with non-CT-containing alfalfa (*Medicago sativa*) forage. Previous in vitro and in vivo research showed that the addition of plant secondary metabolites, such as CT extract (quebracho) and saponin, reduced CH_4_ production by 6% to 40% per unit of DM [36,50,81]. These values are similar to the results with meta-analyses data of Jayanegara et al. [82], who reported a considerable reduction in enteric CH_4_ emissions, but appeared to depend on the source of CT, molecular weight (MW), and dose level of tannins [81]. In vitro studies have also shown that tannins have anti-methanogenic activity, either directly by inhibiting methanogens or indirectly by rumen fermentation profiles (e.g., volatile fatty acids) or microbiota changes (e.g., protozoa, Firmicutes, and Bacteroidetes [29,82,83,84]). However, several studies showed inconsistent results of anti-methanogenesis activity between HT and CT, and/or molecular weight of tannins [10,83,85]. In addition, the ability of CT to bind and precipitate protein is not directly related to the inhibition of CH_4_ production [86], and thus, it appears that the potential of tannins to reduce the methanogen population in the rumen cannot be solely attributed to its ability to bind to methanogens. Plant tannins exist in a wide range of tropical and temperate vegetation [65], are extensively available worldwide, and may be an inexpensive approach for livestock producers to mitigate enteric CH_4_ emissions. This study revealed the potential for using dietary CT to reduce enteric CH_4_ emissions while improving ruminant performance.

## 5. Conclusions

The current results indicate that CT supplementation induced changes in ruminal bacterial community structure and reduced CH_4_ emissions, and increased animal performance. The similarity analysis (% SC) allows the efficient differentiation between microbial ecosystems, and can be used for the fast screening of phylogenic microbiota changes on a given diet in a culture-independent manner. It can be concluded that moderate concentrations of CT (0.05–0.07% CT/kg BW) in diets can be used to improve the efficiency of animal production in grazing ruminants while reducing enteric CH_4_ production. As a result, CT improved ADG in steers grazing winter wheat forage. This is related to the reduction in CH_4_ production and microbiota changes. Therefore, it is important to capitalize on the beneficial effects of moderate levels of CT supplementation, associated with winter wheat forage, by growing stocker cattle in winter seasons. Further in vivo studies are needed to verify the effects of sources of tannins, characterized by molecular weight and structural composition, on reducing methanogenesis and improving feed efficiency in ruminants.

## Figures and Tables

**Figure 1 animals-11-02391-f001:**
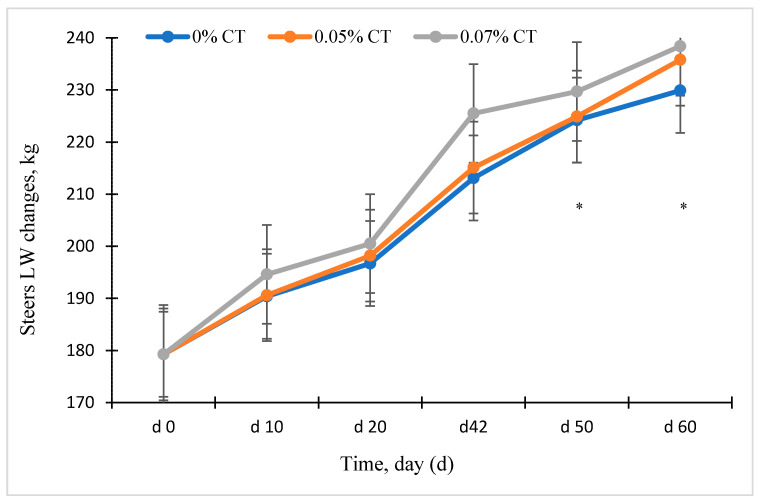
Effects of condensed tannins (CT) on the animal live weight (LW) changes in rumen-cannulated steers grazing winter wheat forage. Animal LW was measured on day (D) 0, 10, 20, 42, 50, and 60. Results are mean values ± SEM (*n* = 6). Initial body weight from d 0 was used as a covariate. Statistical significance: * *p* < 0.05. Values without asterisks are not significantly different (*p* < 0.05).

**Figure 2 animals-11-02391-f002:**
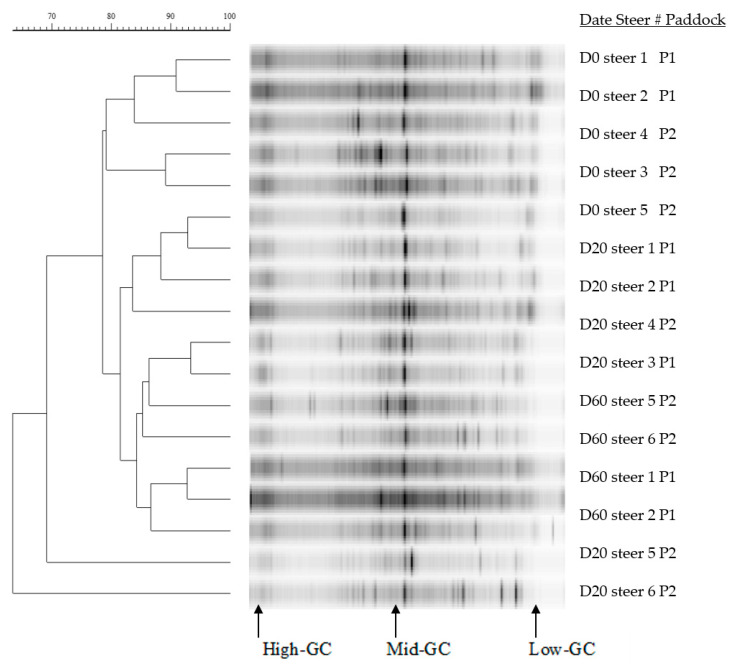
Dendrograms of denaturing gradient gel electrophoresis gels of rumen bacteria 16S rDNA amplicon bands patterns in steers (*n* = 6) given winter wheat forage for day 0, day 20, and day 60 without condensed tannins (CT) supplementation. P1 to P2 = each paddock (P). The bar above the figure indicates Dice percentage similarity coefficients (% SC). D = day, GC= guanines and cytosines. Steer # = tag number (#).

**Figure 3 animals-11-02391-f003:**
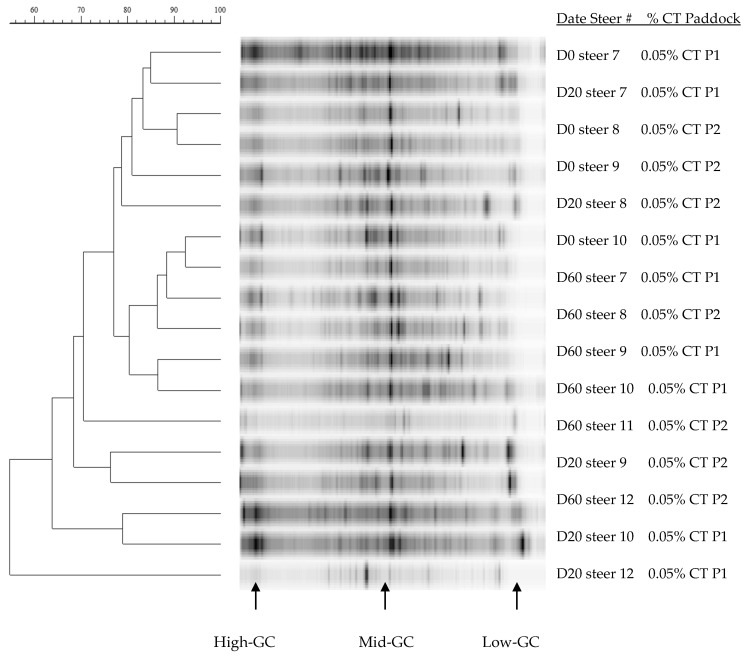
Dendrograms of denaturing gradient gel electrophoresis gels (DGGE) of rumen bacteria 16S rDNA amplicon bands patterns in steers (*n* = 6) given winter wheat forage for day 0, day 20, and day 60 and containing 0.05% condensed tannins (CT; %/kg BW). P1 to P2 = each paddock (P). The bar above the figure indicates Dice percentage similarity coefficients (% SC). D = day, GC= guanines and cytosines, and Steer # = tag number (#).

**Figure 4 animals-11-02391-f004:**
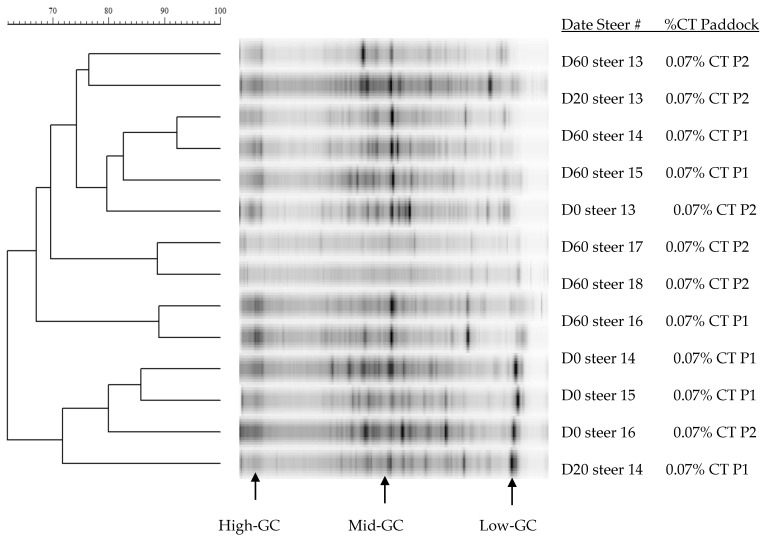
Dendrograms of denaturing gradient gel electrophoresis gels of rumen bacteria 16S rDNA amplicon bands patterns in steers (n = 4 to 5) given winter wheat forage for day 0, day 20, and day 60 and containing 0.07% condensed tannins (CT; %/kg BW). Cattle tag numbers (#) of 17 and 18 were missed on day 0 and day 20. P1 to P2 = each paddock (P). The bar above the figure indicates Dice percentage similarity coefficients (%SC). D = day, GC= guanines and cytosines, and Steer # = tag number (#).

**Table 1 animals-11-02391-t001:** Chemical composition and in vitro dry matter (DM) digestibility (IVDMD) of winter wheat forage diets (*n* =3).

Item	Winter Wheat Forage (DM Basis)			
	7 March	17 April	SEM	*p*-Value
Dry matter	92.2	93.5	1.52	0.15
Crude protein	29.5 ^a^	13.7 ^b^	1.26	0.001
Neutral detergent fiber	42.2 ^b^	47.9 ^a^	0.80	0.01
Acid detergent fiber	26.8	27.5	3.36	0.11
IVDMD, %	90.1 ^a^	85.4 ^b^	1.32	0.001

^a,b^ Means ± SEM in rows and with different superscripts are significantly different (*p* < 0.05). IVDMD = in vitro dry matter (DM) digestibility. SEM = standard error of the means.

**Table 2 animals-11-02391-t002:** Effects of condensed tannins (CT) on the animal body weight (BW) changes and average daily gain (ADG) in rumen-cannulated steers grazing wheat forage.

Item	Percentage of CT/kg BW			SEM	*p*-Value
	0	0.05	0.07		
Number of steers	6	6	6	-	-
Animal performance					
Day 0, before co-variate	170.3	152.3	179.6	7.10	0.21
Day 0, after co-variate ^1^	181.7	181.7	181.7	-	-
Day 10	192.9	192.6	197.1	2.37	0.28
Day 20	199.6	200.7	201.3	2.13	0.34
Day 42	215.1	217.6	226.2	7.04	0.32
Day 50	226.8	227.7	232.3	2.03	0.10
Day 60, final BW	232.3 ^b^	238.4 ^a,b^	246.0 ^a^	2.50	0.04
ADG, kg/d	0.84 ^b^	0.94 ^a,b^	1.00 ^a^	0.045	0.04
*p*-value					
CT	-	-	-	-	0.17
Time	-	-	-	-	0.01
CT by time interaction	-	-	-	-	0.65

BW = body weight, ADG = average daily gain, SEM = standard error of the mean, and - = not determined. ^1^ Initial body weight (BW) from day 0 was used as a covariate. ^a,b^ Means within rows with a different superscript differ significantly, *p* < 0.05.

**Table 3 animals-11-02391-t003:** Effects of condensed tannins (CT) on the rumen fluid DNA concentrations in rumen-cannulated steers grazing wheat forage.

Item	Percentage of CT/kg BW			SEM	*p*-Value
	0	0.05	0.07		
Number of steers	6	6	6	-	-
Ruminal DNA concentration (ng/µL)					
Day 0	27.5	24.9	24.7	3.85	0.87
Day 20	65.4	75.2	80.7	12.51	0.20
Day 60	100.4	112.5	129.3	12.61	0.06
Average	64.4 ^b^	70.8 ^a,b^	78.2 ^a^	9.65	0.04
*p*-value					
CT	-	-	-	-	0.12
Time	-	-	-	-	0.001
CT by time of grazing interaction	-	-	-	-	0.84

BW = body weight, CT = condensed tannins. ^a,b^ Means within a row with a significant superscript differ at *p* < 0.05.

**Table 4 animals-11-02391-t004:** Effects in rumen fluid from steers fed condensed tannins (CT) on in vitro ruminal gas and methane (CH_4_) gas production when incubated with minced fresh wheat forage substrates (harvested from 7 March to 19 April) and mixed rumen microorganisms in each treatment obtained from rumen-cannulated steers grazing wheat forage with CT supplementation.

Item ^1^	Percent of CT/kg BW				
	0	0.05	0.07	SEM	*p*-Value
In vitro gas production					
Total gas, mL/12 h	58.3 ^a^	41.5 ^b^	36.0 ^b^	3.28	0.01
Total gas, mL/g DM	33.7 ^a^	23.9 ^b^	20.9 ^b^	1.52	0.001
Gas production parameters ^1^					
Gas potential (a + b)	72.3 ^a^	52.9 ^b^	49.4 ^b^	5.11	0.01
Gas rate (c) ^2^	0.133	0.107	0.110	0.020	0.39
CH_4_ production					
CH_4_ (mL/g DM)	0.91 ^a^	0.73 ^a,b^	0.26 ^b^	0.09	0.001
CH_4_ (mg/g DM)	0.65 ^a^	0.52 ^a,b^	0.19 ^b^	0.07	0.001

^1^ Mean values (*n* = 3). BW = body weight, SEM = standard error of the mean, DM = Dry matter, and CH_4_ = methane. ^2^ The values of a + b, and c were constants of the exponential equation, where a = gas production at time 0, b = the proportion of gas production during time t, a + b = gas potential, and c = gas rate of the *b* fraction. ^a,b^ Means within a row with a significant superscript differ at *p* < 0.05.

## Data Availability

The data that support the findings of this study are available from the corresponding author upon reasonable request.

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
