# Peer review of "Effects of Condensed Tannins Supplementation on Animal Performance, Phylogenetic Microbial Changes, and In Vitro Methane Emissions in Steers Grazing Winter Wheat"

_animals, 2021, doi:10.3390/ani11082391_

Round 1

Reviewer 1 Report

The effects of tannin on growth performance, microorganism and methane yield of beef cattle were studied. The article has certain meaning for guiding practice. However, the innovation of the paper is relatively low because there are many studies about tannins, as well as the DGGE methods used by the author can not well explain microbiota changes.  Another question is that the trial did not show the inflection point of CT dose response, it is difficult to obtain an appropriate CT doses in terms of animal growth performance and methane production. Other suggestions as below:

It is suggested to simplify the introduction section by shorten the PCR-DGGE and the bacteria reflected by the GC content.

Please spcify the content of SC used in the trial. most places are 0,0.5%,0.75% /kg of BW, but the fig.2 and 3, and line 282 are 1% and 2%. 

L148,149, worm should be warm?

L161: 12-d should be 12-h?

the forage was harvested twice at march 7 and april 17 in line 125, but it is from march 7 to april 19.  please check it.

Table 1 shoud supplement dry matter of the wheat.

Is the ADG value of the 0.07 group right in table 2? 

the author mentioned in the statistial saction that the main effect means of treatment and time were given, but I did no find them in the table2 and 3.

Author Response

REVIEWER 1 – Minor

The effects of tannin on growth performance, microorganism and methane yield of beef cattle were studied. The article has certain meaning for guiding practice. However, the innovation of the paper is relatively low because there are many studies about tannins, as well as the DGGE methods used by the author can not well explain microbiota changes.  Another question is that the trial did not show the inflection point of CT dose response, it is difficult to obtain an appropriate CT doses in terms of animal growth performance and methane production. Other suggestions as below:

It is suggested to simplify the introduction section by shorten the PCR-DGGE and the bacteria reflected by the GC content.

Thank you for your comment. It has been replaced or revised.

Please spcify the content of SC used in the trial. most places are 0,0.5%,0.75% /kg of BW, but the fig.2 and 3, and line 282 are 1% and 2%. 

Changed to 0, 0.05 and 0.07%.

L148,149, worm should be warm?

Changed to “Warm water”

L161: 12-d should be 12-h?

Changed to 12-h

the forage was harvested twice at march 7 and april 17 in line 125, but it is from march 7 to april 19.  please check it.

Corrected to March 07 and April 17.

Table 1 shoud supplement dry matter of the wheat.

Added dry matter content of wheat forage

Is the ADG value of the 0.07 group right in table 2? 

Yes, this is ADG value,

the author mentioned in the statistial saction that the main effect means of treatment and time were given, but I did no find them in the table2 and 3.

The added time and time x treatment interactions.

Reviewer 2 Report

the presented work is an interesting study about the use of Condense Tammins to reduce methane without compromising animal performance. However there are some questions to be addressed prior any consideration for its publication in Animals Journal.

The introduction is well conducted but justification of work is a bit poor, since there is extensive literature of the use of CT in ruminants, maybe not especifically in wheat grazing but there are plenty of similar published works. I recommend authors not to be so cathegorical when they stated "Despite years of research on CT supplementation in ruminant diets, the practical (e.g., on-farm) impacts of supplementation of CT on animal performance, rumen microbiome changes, and enteric methane (CH4) emission are still poorly understood in stocker cattle grazing winter wheat forage", especially because the trial may not be able to resolve all the questions arised.

Then the methods are a bit messy, there are few aspects not fully described, starting for the CT dosage. When Authors indicate 0.05 or 0.07% of CT per kg BW, are they using a mean BW at the beginning of the experiment, or one individual per each animal? Is this level changing over time, since animals still growing? Authors must explain better which is the exact amount given to the animals over time, and whether is constant or not; anyway it is surprising why authors didn´t use a better way to fix the dose as a % to either DM intake or CP intake, as any other feed additive. Please justify it.

I praise the Authors in keeping using DGGE as a way to study microbial structure having other methodolgies much more powerful to get to that point, but I have the impression that they could have done a bit more in this sense. Why did authors load separate gels per treatment? as it is now there is low or no chance to compare any treatment with the control, and hence prove that teh GC content were changing with the use of CT. I feel this is a major flaw and should be justified accordingly.

Moreover, authors claim differences in GC content between Bacteroidetes and Firmicutes, but in L74-75 both belong to por GC content (46.0 and 43.1% respectively). PLease, comment this, since it is a major issue in order to sustain some of the discussion showed in the manuscript. Anyway, discussion is too speculative and highly dependant in others studies based on other technologies (such as NGS) and hence not directly comparable. I recommend authors be careful and limit the discussion.

Finally, althouh the in vitro study is of little originality, rounds off the study, and justify the discussion related with GHG...however I missed some methane measurements in vivo, although I understand maybe the capabilities of the research group did not allow it.

Minor comments:

In L74 and L80 authors contradict themselves stating both a high and low GC content in Bacteroidetes phyla

L148 warm

L149 pre-warmed

L155 which particle size?

L216 Seeing the results I wonder why authors didn´t use a repeated measured design, that should apply a Mixed model in SAS. Please justify it or amend it.

L230 the formula missed the superscript and can lead to a mistake; this is the right one:

Y = a + b(1-e-ct). In general, the use of both superscripts and subscripts is very poor in the manuscript

L257 I suggest to transform table 2, that has a quite confusing structure, into a Figure.

Table 4: “gas potential” and “gas rate” instead of “potential gas” and “rate of gas”.

Author Response

REVIEWER 2 - Major

The presented work is an interesting study about the use of Condense Tannins to reduce methane without compromising animal performance. However there are some questions to be addressed prior any consideration for its publication in Animals Journal.

The introduction is well conducted but justification of work is a bit poor, since there is extensive literature of the use of CT in ruminants, maybe not especifically in wheat grazing but there are plenty of similar published works. I recommend authors not to be so cathegorical when they stated "Despite years of research on CT supplementation in ruminant diets, the practical (e.g., on-farm) impacts of supplementation of CT on animal performance, rumen microbiome changes, and enteric methane (CH4) emission are still poorly understood in stocker cattle grazing winter wheat forage", especially because the trial may not be able to resolve all the questions arised.

Answer: Thank you for your comment. It has been replaced.

“Despite years of research on CT supplementation in ruminant diets, the practical (e.g., on-farm) impacts of supplementation of CT on animal performance, rumen microbiome changes, and enteric methane (CH4) emission are still poorly understood in stocker cattle grazing winter wheat forage, especially because the trial may not be able to resolve all the questions arises”.

Then the methods are a bit messy, there are few aspects not fully described, starting for the CT dosage. When Authors indicate 0.05 or 0.07% of CT per kg BW, are they using a mean BW at the beginning of the experiment, or one individual per each animal? Is this level changing over time, since animals still growing?

Answer: Thank you for your comment. It has been replaced.

 “The amount of CT extract supplementations was adjusted by changing over time depending on an individual animal’s BW growth”.

 Authors must explain better which is the exact amount given to the animals over time, and whether is constant or not; anyway it is surprising why authors didn´t use a better way to fix the dose as a % to either DM intake or CP intake, as any other feed additive. Please justify it.

Answer: Sorry for confusing. We agree to reviewer’s comments. However, on the grazing steers, it is not an easy task to measure dry matter intake in our laboratory capacity (e.g., man-powers and facility). However, dry matter intake is highly associated with BW. So, we decided to feed animals depending on the animal’s BW growth. 

I praise the Authors in keeping using DGGE as a way to study microbial structure having other methodolgies much more powerful to get to that point, but I have the impression that they could have done a bit more in this sense. Why did authors load separate gels per treatment? as it is now there is low or no chance to compare any treatment with the control, and hence prove that teh GC content were changing with the use of CT. I feel this is a major flaw and should be justified accordingly.

Answer: It is a good comment, but 18 steers x 3 treatments are a large set to visualized and hard to see individual animal variations. 

Moreover, authors claim differences in GC content between Bacteroidetes and Firmicutes, but in L74-75 both belong to por GC content (46.0 and 43.1% respectively). PLease, comment this, since it is a major issue in order to sustain some of the discussion showed in the manuscript. Anyway, discussion is too speculative and highly dependant in others studies based on other technologies (such as NGS) and hence not directly comparable. I recommend authors be careful and limit the discussion.

Answer: Thank you for the comments: revised the sentence.

Finally, althouh the in vitro study is of little originality, rounds off the study, and justify the discussion related with GHG...however I missed some methane measurements in vivo, although I understand maybe the capabilities of the research group did not allow it.

Answer: It is a good comment. Yes, we do not have any chambers or Greenfeed systems. So, we decided to measure methane emissions using an in vitro system only.

Minor comments:

In L74 and L80 authors contradict themselves stating both a high and low GC content in Bacteroidetes phyla.

Answer: Revised this sentence to “A phylum of Firmicutes, a Gram-positive bacteria that is grouped based on its low-low GC-content [22, 25, 26]. In contrast, the phylum of Bacteroidetes is composed of bacteria that are generally Gram-negative, with a high GC content [27]”

L148 warm . Changed to “Warm”  changed to “warm”

L149 pre-warmed: Changed to “pre-warmed”

L155 which particle size? Added “2-3 mm particle size”

L216 Seeing the results I wonder why authors didn´t use a repeated measured design, that should apply a Mixed model in SAS. Please justify it or amend it. Answer: One of our statistic expertise (he is now retired) has recommended using GLM procedure due to no time x treatment effects.

L230 the formula missed the superscript and can lead to a mistake; this is the right one:

Y = a + b(1-e-ct). In general, the use of both superscripts and subscripts is very poor in the manuscript.

Changed to “Y = a + b(1-e-ct)”.

L257 I suggest to transform table 2, that has a quite confusing structure, into a Figure. Added Figure for body weight changes in Figure 1.

Table 4: “gas potential” and “gas rate” instead of “potential gas” and “rate of gas”.

Changed to “gas potential” and “gas rate”

Round 2

Reviewer 2 Report

Authors have made an effort in improving the manuscript. There are only a few minor amendments to be conducted prior its acceptance.

First, authors must explain in a comprehensive way CT administration. In L136-137 they wrote "The amount of CT extract supplementations was adjusted by changing over time depending on an individual animal’s BW growth". Authors should explain the way of adminitration, and the frequency (daily, weekly). I guess animals were weighed before administration in order to get precise recording of animals BW, but again, this is not explained in the text.

I also recomend to rephrase sentence in L88-L92, honestly, made changes haven´t improved the text.

Author Response

Dear Sir/madam

Thank you so much for your great help.

I changed sentences according to your suggestions:

1) L134-137 (L159-164 now):  Two levels of quebracho CT (0.05% and 0.07% CT/kg of BW) and control (0% CT; warm water only) treatment were premixed with warm water (approximately 30°C) and introduced once daily (0800) via rumen cannula of steers grazing wheat forage as a main source of diet. The amount of CT extract supplementations was adjusted by changing over time depending on an individual animal’s BW growth. Animals were weighed at 10 days intervals before administration in order to get a precise recording of the animal’s BW. 

2) L88-92: deleted a sentence.

Thank you so much
